# OpenReview forum: "Mitigating Reward Overoptimization via Lightweight Uncertainty Estimation"
_NeurIPS.cc/2024/Conference — NeurIPS 2024 poster_

### Official Review · Reviewer_fmPe · 2024-06-20

**Soundness:** 2
**Presentation:** 2
**Contribution:** 2
**Rating:** 6
**Confidence:** 4

**Summary:**

Quantifying the uncertainty of the reward model output can mitigate the issue of reward over-optimization. This paper  introduces a lightweight method for quantifying reward uncertainty in RLHF, which can be  integrated into existing trained reward models.
Then the authors propose a distributionally robust optimization procedure to counter overoptimization during policy improvement.
Experiment results verify the proposed method.

**Strengths:**

1. The paper is generally easy-to-follow.
2. Uncertainty estimation is an important topic in LLM and worth specialized study.
3. The proposed method perform well in experiments.

**Weaknesses:**

1. The proposed method requires calculating matrix inverse $M_D^{-1}$, which can be numerically unstable and time consuming when the dataset size $N$ is big.
2. The theoretically motivation (Theorem 3.1) has substantial gap with the proposed practical method, e.g., requiring infinitely wide neural network.
3. The proposed uncertainty measure $U^{CI}$ has no statistically justification/guarantee and seems to only upper bound the difference between the true reward and the learned reward, as seen in Eq. (4). It is unclear if $U^{CI}_{x,y}$ can measure the "uncertainty" of the estimated reward. And if so, does it measure epistemic or aleatoric uncertainty? And is it possible to construct any statistically valid confidence set based on the proposed uncertainty measure?
4. The proposed method requires reference responses, which is a non-standard and stronger assumption for RLHF's reward learning and policy optimization. This casts doubts on the practical value of the proposed method and the fairness of the experimental comparison.

**Questions:**

1. What is the definition of "uncertainty" that this paper centers around?
2. Why will the uncertainty measure $U^{CI}$ be smaller when $(x,y)$ is close to the training data samples and $U^{CI}_{x,y}$ be higher when $(x,y)$ is far from the training data?
3. L185-186, how is $C^r_{\delta}$ be constructed such the ground truth reward is included with probability of $1-\delta$ ?

**Limitations:**

The authors have addressed the limitations of the paper.

---

> ### Author Rebuttal · Authors · 2024-08-07
>
> # Reply to Reviewer fmPe
>
> > [Q1] "Clarification on uncertainty: (1) definition of uncertainty in this paper, epistemic or aleatoric uncertainty?  (2) The proposed uncertainty measure has no statistically justification/guarantee and seems to only upper bound the difference between the true reward and the learned reward, as seen in Eq.4. (3) Assumption in Theorem 3.1.; (4) Calculating $M_D$ can be numerically unstable and time-consuming when the dataset size is big."
>
> **For (1)**, in this paper, we focus on quantifying the **epistemic uncertainty** of the estimated reward from the learned reward model. Formally, assuming the reward model is parameterized by $\varphi$. For a  prompt-response pair $(x,y)$, let $r\_{\hat{\varphi}}(x,y)$ denote the estimated reward, and $r^*(x,y)$ denote the groundtruth reward under the optimal parameterization $\varphi^*$.   Then the reward uncertainty $U^{\delta}\_{x,y}$ implies that with probability $1-\delta$, the inequality
> $$|r\_{\hat{\varphi}}(x,y) - r^*(x,y)| \leq U^{\delta}\_{x,y}$$
> holds.
> In other words, with probability  $1-\delta$:
>  $$r^*(x,y) \in C^\delta\_{x,y} :=  [r\_{\hat{\varphi}}(x,y)-U^{\delta}\_{x,y}, r\_{\hat{\varphi}}(x,y)+U^{\delta}\_{x,y}]$$
> where $C^\delta_{x,y}$ is called the confidence interval of the $r_{\hat{\varphi}}(x,y)$.
>
>
> **For (2)**, we thank the reviewer for pointing out an area that may have caused unnecessary misunderstanding.   As discussed above and in Theorem 3.1 (lines 151-153), the uncertainty indeed has a statistical justification: it holds with probability $1-\delta$, and the uncertainty value is determined by $\delta$ (typically the smaller $\delta$ is, the larger uncertainty value will be).
>
> **For (3)**, we agree with the reviewer that Theorem 3.1 relies on certain assumptions regarding network architectures, specifically that the network width is infinitely wide. Such an assumption is commonly adopted in analyses involving the Neural Tangent Kernel, as in [1].
>
> This is also why we empirically examined the effectiveness of the proposed lightweight uncertainty estimation using a synthetic setup with known ground-truth rewards in Section 5.1 (Figure 1). The results demonstrate that the proposed lightweight uncertainty estimation can accurately capture the divergence between the ground truth and estimated proxy rewards, effectively signalling over-optimization.
>
>
> **For (4)**, we thank the reviewer for pointing out this typo.  The correct formula for calculating $M_D$ is $M_D = \lambda I + \frac{1}{N}\sum_{i=1}^N \sum_{y \in [y_c^i, y_r^i]}e(x_i, y)e(x_i,y)^T$.  With $\lambda >0$, $M_D$ is a positive definite matrix, ensuring the existence of its inverse.
>
> Moreover, calculating $M_D$ requires only a single pass through the reward training data. This computational cost is significantly lower compared to querying each reward ensemble for every sample during RL training, as done by ensemble-based methods.
> In addition, the memory cost of maintaining $M_D$ is much lower than keeping multiple reward models in memory.
>
> [1]. Sadhika Malladi, Alexander Wettig, Dingli Yu, Danqi Chen, and Sanjeev Arora. A kernel-based view of language model fine-tuning. ICML 2023.
>
> > [Q2] "Why will the uncertainty measure be smaller when (x,y) is close to the training data samples and be higher when (x,y) is far from the training data?"
>
> Recall that for a  prompt-response pair $(x,y)$, its uncertainty is calculated through
> $$U_{x,y}^{CI}(x,y) = b\sqrt{e(x,y)^T M_D^{-1} e(x,y) },$$
> where $e(x,y)$ denotes the last-layer embedding for the pair.
>
> The term $\sqrt{e(x,y)^T M_D^{-1} e(x,y) }$ represents the Mahalanobis distance,  measures the distance of a point $(x,y)$  from the distribution of the training data captured by $M_D$.  Thus if the pair $(x,y)$ is far from the training data samples, it will result in a longer distance and, consequently, a higher uncertainty value.
>
> > [Q3] "Clarification on Reference Responses."
>
> The reference response can be any reasonably good answer, as long as it achieves a positive reward on average. In our experiments, we found that even when sampling responses from the SFT policy, this requirement can be met. Therefore, it does not introduce additional burdens in practical scenarios.
>
> Moreover, if high-quality reference responses are used, such as annotated good responses from users or responses generated by a well-performing model, the performance of AdvPO can be further enhanced, as demonstrated in the TLDR dataset in Table 1. This is because the inclusion of reference responses prevents AdvPO from being overly or wrongly pessimistic by guiding policy optimization towards the direction of the reference responses while optimizing against pessimistic rewards, as shown by Lemma D.1.
>
> For a fair comparison, we have incorporated PPO-ref, a modified version of PPO that includes reference responses, as one of our baselines. Table 1 demonstrates the benefits of AdvPO over PPO-ref through addressing over-optimization. Additionally, we also compare AdvPO with ensemble-based methods that use reference responses. Further details can be found in the response to CQ1.
>
> >[Q4] "In L185-186, how is $C_{\delta}^r$ constructed?"
>
> We cannot construct $C_{\delta}^r$ in practical scenarios with large parameter sizes. However, the success of the lightweight uncertainty estimation and the theoretical insight from Theorem 3.1 implies that the uncertainty primarily stems from the inaccuracy in the estimated projection weights.
> Thus, AdvPO opts for a relaxation that minimizes an upper bound (lines 195-204), resulting in the objective in Eq.(5), where the minimization is now taken over the projection weights instead of the reward functions
> Experimental results in Sections 5.2 and 5.3 show the effectiveness of AdvPO, even with this relaxation.
>
> We hope that the above has addressed all the reviewer's concerns and that the reviewer would consider raising their score. We are happy to answer any additional questions the reviewer might have.

---

> > ### Comment · Reviewer_fmPe · 2024-08-10
> > **Response to the authors**
> >
> > Dear authors,
> >
> > Thank you so much for your detailed responses. I have increased my rating to 6.

---

> > > ### Author Response · Authors · 2024-08-12
> > > **Reply to the response.**
> > >
> > > Thank you for your positive feedback. If you have any further questions or need additional clarification, please feel free to ask. We are more than happy to engage in further discussion.

---

### Official Review · Reviewer_NjZD · 2024-06-26

**Soundness:** 2
**Presentation:** 3
**Contribution:** 3
**Rating:** 6
**Confidence:** 3

**Summary:**

This paper aims to tackle the problem of reward model overoptimisation in RLHF. To do this, they propose a new method for quantifying the uncertainty of the reward model on a given input, and penalise the reward of the policy during RLHF training based on this uncertainty estimation. The uncertainty estimation is calculated from the final layer embeddings of the reward model, and the authors provide several theoretical results to show that the adversarial objective they initially propose can be optimised with standard RL algorithms. Empirically, they show in simulated settings that their method performs better than mean ensembles and lora ensembles on both Anthropic HH and TL;DR datasets in terms of final performance and qualitatively mitigating overoptimisation. Several ablation studies are performed to demonstrate the importance of each of the components of the proposed algorithm.

**Strengths:**

The method the paper introduces is interesting and novel, and the benefit of lightweight uncertainty estimation vs training and holding in GPU memory an entire ensemble is substantial. While using uncertainty to address overoptimisation is not novel, this approach to quantifying uncertainty is.

The theoretical analysis in the work seems sound and gives insight and intuition about how and why their method is effective.

The presentation and clarity of the paper is good, and the paper is generally clear and easy to read.

The problem the paper tackles is important, and while a variety of work currently exists in this space, there is not yet a clear solution to the problem of overoptimisation, so this work is useful and significant in this regard.

**Weaknesses:**

### Insufficient comparison to baselines

In general, the comparison to baselines in the settings you consider isn't sufficient to support the claims that the method is outperforming existing SOTA. It would be beneficial to compare to WARM from https://arxiv.org/abs/2401.12187 (or a variant of it that just uses the ensemble models), and UWO from Coste et al. This is specifically about the results in figure 2 and table 1.

I think it's important to compare against an ensemble of RMs which are all the same size as the RM used by AdvPO. While this doesn't normalise the model sizes between the ensemble and the single RM, often the limiting factor here will be how good the largest pretrained model you have access to, which means comparing to an ensemble of RMs with the same size as the RM used for AdvPO makes sense, although AdvPO wouldn't need to beat this baseline, but it would be useful to see the comparison.

### Unclear simulated setting

In the results of section 5.2, the gap between the trained and gold reward model is 7B -> 13B, which is quite small - Most previous works have had a gap of at least 1.4B -> 7B, and often much larger. This makes the setting less analogous to the setting with real human preferences. It would be beneficial to do these experiments with smaller trained RMs and policies but the same gold RM

### Limited model sizes

While it's an easy comment to make, it is the case that the results are only for one policy and RM size in both sections. It would be beneficial to have results on additional model sizes (either smaller or larger), to ensure the method works robustly across scales

## Summary

Overall I'm currently giving the paper a borderline accept (5). I think the promise of the method and results outweigh potential issues with the experimental setting and baseline comparison currently. I'd be willing to raise my score to a 6 or 7 if comparisons to more baselines were made, and experiments with different model sizes were performed.

[EDIT]: I have raised my score to a 6 given the baseline being compared to was stronger than I previously thought.

**Questions:**

- In figure 1 c, why does ENS-3B start much higher than ENS-7B and CI?
- What is the ENS-s method exactly? are you using the variance of the ensemble, or something else? How do you choose the hyperparameter for the weight of this variance.

**Limitations:**

The authors discuss the limitations of the work somewhat, but I would appreciate more discussion of the limitations of the experimental setup (smaller models, gpt-4 evaluation, limited realism of datasets).

---

> ### Author Rebuttal · Authors · 2024-08-07
>
> # Reply to Reviewer NjZD
>
> First, we would like to thank the reviewer for the comprehensive review of our paper and for acknowledging the novelty and benefits of our proposed AdvPO. We have carefully read through your review and added corresponding experiments. We hope the following clarifies any misunderstandings.
>
> > [Q1] "Comparisons to baselines: (1) comparison to ensemble-based baselines. (2) What is the ENS-s method exactly? (3)compare against an ensemble of RMs which are all the same size as the RM used by AdvPO.  "
>
> **For (1)**, we would like to clarify a misunderstanding. First of all,  we have already included ensemble models (UWO from Coste et al.) in our experiments on both the uncertainty analysis in Figure 1 (ENS-3B and ENS-7B) and the resulting policy in Table 1 ("ENS-s" in Table 1).
>
> In Figure 1, we found that the proposed lightweight uncertainty (CI) surpasses reward ensembles with comparable parameter sizes—ENS-3B, which ensembles 3x3B reward models.
>
> In Table 1, we also observe that AdvPO consistently outperforms ENS-s, which ensembles 3x3B reward models for reward and uncertainty calculation, on both datasets. It's worth noting that this ensemble has more parameters than the reward model we used (7B), and hence we believe that this is indeed a fair comparison.
>
> Additionally, we performed further evaluation on ENS-s in line with the experimental framework depicted in Figure 2. The results are shown in Figure 2 in the attached PDF.
>
> We can observe that on the Anthropic HH dataset, with reliable uncertainty estimation as shown in Figure 1, ENS-s helps mitigate overoptimization, i.e., the gold rewards do not decrease as PPO optimization progresses. Although we found it needs a higher KL penalty than AdvPO, the performance is not as effective as AdvPO.
> However, on the TLDR dataset, without reliable uncertainty estimation, even with a higher KL penalty, the performance of ENS-s is significantly worse.
>
>  **For (2)**, our implementation of ENS-s strictly follows the UWO implementation from Coste et al [1], using three 3B reward ensembles. The reward during PPO training is computed in according to Eq. (3) in the paper (Eq.(5) in [1]). Specifically, we compute the average value from three reward models and subtract their uncertainty, which is determined as the variance of rewards from different reward models.
>
>  The weight of this variance is treated as a hyper-parameter, which is searched within the range [0.01, 0.05, 0.1, 0.5, 1]. We then select the hyper-parameter that achieves the highest reward on the validation set.
>
> **For (3)**, we tried to run ensembles with three 7B reward models. However, we encountered out-of-memory (OOM) issues, since this requires keeping six 7B models in memory (one policy model, one reference model, one critic model, and three rewards models).
>
> We would also like to thank the reviewer for pointing out WARM paper. WARM is still based on the idea of reward ensembles, but averaging them in the weight space. However, this approach still requires additional reward training and tries to perform **mean optimization**, i.e., optimizing towards average rewards from reward ensembles **without considering uncertainties**.
>
> We will incorporate these discussions into the final version of our paper.
>
> [1] Coste, Thomas, et al. "Reward Model Ensembles Help Mitigate Overoptimization." The Twelfth International Conference on Learning Representations. 2023.
>
> > [Q2] "Unclear simulated setting & Limited model sizes: Experiments with smaller trained RMs and policies but the same gold RM."
>
> We thank the reviewers for their valuable suggestions. In response, we have conducted experiments using 3B policy models and 3B reward models, while maintaining the same gold reward model as the reviewer suggested.
> Due to time constraints, we primarily focused on exploring the effectiveness of lightweight uncertainty estimation with both the policy model and reward models initialized from a 3B model (OpenLLaMA3B). Once we achieve a reliable uncertainty estimation, leveraging such uncertainties to mitigate overoptimization becomes straightforward.
>
> The experimental setup follows Section 5.1, with the only difference being the model size. Besides the proposed lightweight uncertainty method (CI), we also employ ENS-3B, which uses three 3B models thus requiring (3x) the computational resources,  for reference.
>
> The experimental results are demonstrated in Figure 3 in the attached PDF. We can observe from Figures 3b and 3d that on both datasets, as the difference between gold and proxy rewards increases, the uncertainty calculated by our CI also rises, indicating the reliability of the uncertainty estimation method. Moreover, similarly to the experiments on 7B models, we can observe in Figures 3a and 3c that CI remains effective in signalling the divergence of gold and proxy rewards.
>
> > [Q3]"In figure 1 c, why does ENS-3B start much higher than ENS-7B and CI?"
>
> We thank the reviewer for the detailed review and for pointing out some parts that caused unnecessary misunderstanding.
> Different uncertainty estimation methods have different scales, so for ease of illustration, when plotting Figure 1, we rescaled the uncertainties of each method to the [0,1] range using min-max scaling. As a result, ENS-3B starts much higher than ENS-7B and CI suggests that its estimated uncertainty is initially higher but decreases as the PPO optimization steps progress (maybe because the model collapses as samples move further away from the original data and hence the variance decreases).
>
> This trend implies that the uncertainty estimates are not reliable, as over-optimization leads to divergence between the proxy reward and the golden reward (i.e., the uncertainty should be increased).
>
> We hope that the above has addressed all the reviewers concerns and that the reviewer would consider raising their score.

---

> > ### Comment · Reviewer_NjZD · 2024-08-10
> > **Response**
> >
> > Thank you for your response and clarification.
> >
> > It is very unclear in the paper currently that you compare against UWO from Coste et al. as opposed to the mean ensemble or WCO. It would be beneficial to make that clearer, as well as the discussion of hyperparameter choice. I think it would also be beneficial to compare against the other methods there as well as just UWO.
> >
> > > [on WARM] However, this approach still requires additional reward training and tries to perform mean optimization, i.e., optimizing towards average rewards from reward ensembles without considering uncertainties
> >
> > I agree that conceptually it may be different to your method, but I believe it is still a necessary baseline to compare against, given it is tackling the same problem of overoptimisation.
> >
> > Thanks for the rest of your response and clarification, it was very helpful. I am happy to raise my score to a 6, given the baseline being compared against is better than I thought (I believed it was mean ensemble, but it is actually UWO). I would raise my score higher if there were more baseline comparisons or results for the 7B ensemble reward models (although I realise those are unlikely in the remaining time).

---

> ### Author Response · Authors · 2024-08-12
> **Reply to the response**
>
> We thank the reviewer for the instant feedback and valuable suggestions. We will revise the paper to make it much clearer that we compare against UWO from Coste et.al [1] and discuss the hyperparameters in detail.
>
> We want to further explain the selection of baselines. We only compared to UWO since, as observed in [1], UWO works the best compared to WCO or mean ensembles. We also thank the reviewer for pointing out the paper WARM. As previously discussed, WARM aims to learn a robust reward model through a weighted average of multiple reward models. We want to further remark that this is orthogonal to our methods. Our methods can be plugged into any learned reward model to calculate uncertainties and leverage uncertainty for better policy optimization under the reward model.
>
> Moreover, following the reviewer's strong suggestion to compare with 7B reward ensembles, we are trying to run experiments with 7B reward ensembles, and we will post the results when available. (Although this causes a significant computational cost with 3x7B reward models, while our method only employs one 7B model, as the reviewer remarks that AdvPO wouldn't need to beat this baseline.)
>
> [1] Coste, Thomas, et al. "Reward Model Ensembles Help Mitigate Overoptimization." ICLR 2023.

---

> > ### Author Response · Authors · 2024-08-14
> > **Results on comparison with 7B ensembles.**
> >
> > We thank the reviewer for the valuable suggestions. In response,  we compare AdvPO with ENS-7B, which uses three 7B reward ensembles (as defined in Eq.(3) in the paper). Note that AdvPO only requires one 7B reward model. The weight of uncertainty in Eq.(3) is still treated as a hyperparameter and searched within the range [0.01, 0.05, 0.1, 0.5, 1]. We then select the hyperparameter that achieves the highest reward on the validation set.
> >
> > The results are shown in the following table.
> > |                         | Anthropic HH | Anthropic HH | Anthropic HH | Anthropic HH    | TLDR   | TLDR   | TLDR  | TLDR            |
> > |-------------------------|--------------|--------------|--------------|-----------------|--------|--------|-------|-----------------|
> > |                         | Win          | Tie          | Lose         | $\Delta$        | Win    | Tie    | Lose  | $\Delta$        |
> > | AdvPO v.s ENS-7B|       29.3%        | 48.8%        | 21.9%        |  $\uparrow$ 7.4          | 60%    | 7%     | 33%   |  $\uparrow$ 27             |
> > | AdvPO v.s ENS-s(ENS-3B) | 43.0%        | 26.5%        | 30.5%        | $\uparrow$ 12.5 | 77%  | 3%  | 20% | $\uparrow$  57 |
> >
> > We can observe that with large reward ensembles, the performance of ENS-7B is much closer to AdvPO. However, AdvPO still achieves better performance than ENS-7B, especially on the TLDR dataset with good reference responses.

---

### Official Review · Reviewer_2q4b · 2024-07-08

**Soundness:** 3
**Presentation:** 3
**Contribution:** 3
**Rating:** 6
**Confidence:** 3

**Summary:**

This paper introduces uncertainty-based methods to tackle the over-optimization issue in RLHF. Drawing inspiration from neural bandits, the authors first propose a lightweight uncertainty estimator based on the final embedding layer. They then formulate the problem as an adversarial optimization task. Empirical experiments are conducted using the Anthropic HH and TL;DR datasets.

**Strengths:**

This paper presents a lightweight method for uncertainty quantification of point-wise rewards, which reduces memory usage compared to standard ensemble-based approaches. Additionally, experimental results demonstrate its effectiveness in mitigating over-optimization issues at the 3B and 7B scale.

**Weaknesses:**

1. In the experiment parts Table 1, ensemble-based baselines do not Incorporate Reference Responses, which leads to the ablation study being less completed.

2. The selection of the last embedding layer appears ad hoc and requires further analysis to justify the choice of different layers.

**Questions:**

1. In Table 1, ENS-s achieves similar performance to PPO in both the Anthropic HH and TL;DR datasets, which seems to contradict the results shown in Figure 1, where ENS-3B appears to help mitigate over-optimization. Could the authors provide more clarification regarding this discrepancy?

2. I observed that LoraEns performs significantly worse than other methods. Is training a LoraEns reward model more challenging than training a standard reward model, thereby making it less accurate? Providing additional results on reward model accuracy or variance, particularly for ensemble-based models, would be helpful.

**Limitations:**

See Weaknesses

---

> ### Author Rebuttal · Authors · 2024-08-07
>
> # Reply to Reviewer 2q4b
>
> We would like to thank the reviewer for the thought-provoking questions to improve our manuscript. We would also like to thank the reviewer for acknowledging that our “experimental results demonstrate its effectiveness in mitigating over-optimization issues at the 3B and 7B scale”. Here below, we will address the reviewer's questions and clarify any misunderstandings.
>
> > [Q1] "In Table 1, ensemble-based baselines do not Incorporate Reference Responses."
>
> Please refer to our Common Response [CQ1].
>
> > [Q2] "The selection of the last embedding layer appears ad hoc and requires further analysis to justify the choice of different layers."
>
> We thank the reviewer for asking this clarification question. Our work is the first to investigate leveraging the internal representation of the reward model for uncertainty quantification. We focus on the last layer due to:
> 1. The reward is acquired through direct projection of last layer embeddings.
> 2. As discussed in Section 3.1, previous work suggests that the last layer embedding captures generalized and rich information. Specifically, [1,2] show that freezing the network up to its last layer and retraining only the projection head with a smaller dataset, free of spurious correlations, improves robustness. [3] indicates that even fine-tuning an LLM's last layer embedding with noisy labels can yield high performance in subsequent classification tasks when the projection weight is accurately derived from ground-truth labels.
> 3. Using only the last layer embedding makes our methods easy to compute and scalable.
>
> Thus, we focused on the last layer in our project and have demonstrated its effectiveness.
>
> We also appreciate the reviewer for highlighting this interesting future direction. We will explore the potential of other intermediate layers for more accurate estimates in future work.
>
> [1] Kirichenko P, Izmailov P, Wilson A G. Last Layer Re-Training is Sufficient for Robustness to Spurious Correlations. ICLR 2023
>
> [2] LaBonte T, Muthukumar V, Kumar A. Towards last-layer retraining for group robustness with fewer annotations[J]. NeurIPS 2023
>
> [3] Burns, Collin, et al. "Weak-to-strong generalization: Eliciting strong capabilities with weak supervision."
>
> > [Q3] "Clarification on the performance of ENS-s and PPO in Table 1."
>
> We first want to clarify that the results in Table 1 are based on pairwise comparisons; for each prompt, we directly compare responses generated by two models. Thus, a response from AdvPO might be better than those from both PPO and ENS-s, while ENS-s's response could still be better than PPO's.
>
> We would also like to clarify that there is no contradiction between the results in Table 1 and Figure 1. For example, in Table 1, on the Anthropic HH dataset, ENS-s has a winning rate of 30.5% against AdvPO, while PPO has a winning rate of 20% against AdvPO, indicating the benefits of ENS-s over PPO.
>
> We further conducted a direct comparison between ENS-s and PPO on Anthropic HH dataset. The win-tie-lose rate is as follows:
>
> |             | Anthropic HH  | Anthropic HH  | Anthropic HH  | Anthropic HH
> |-------------|---------------|---------------|---------------|---------------|
> |             | **Win**           | **Tie**           | **Lose**          | $\Delta$
> | Ens-s v.s PPO  | 37.5%         | 29%        |     33.5 %   | 4%
>
> One can observe that the ENS-s performed slightly better than PPO, aligning with the observation that ENS-3B appears to help mitigate over-optimization on Anthropic HH dataset in Figure 1.
>
> Here, we only report the direct comparisons on the Anthropic HH dataset, since as we discussed in Section 5.3, we adopted a mixed evaluation strategy with human annotation involved. Considering the high cost of human annotation, and the observations that ENS-3B in the TLDR dataset in Figure 1 is not reliable, we only reported direct comparisons on the Anthropic HH dataset.
>
> > [Q4] "Performance of Lora ensembles, and the accuracy and variance of reward models."
>
> The accuracy and variance of fully finetuned 7B reward models, LoRA-based reward ensembles from 7B models, and fully finetuned 3B reward models are as follows:
> |   |                                  | Number of ensembles | Anthropic HH (mean acc) | Anthropic HH (std) | TLDR(mean acc) | TLDR (std) |   |
> |---|----------------------------------|---------------------|-------------------------|--------------------|----------------|------------|---|
> |   | Fully Finetuned 7B reward models | 1                   | 0.7137                  | -                  | 0.667          | -          |   |
> |   | Lora-based 7B reward models      | 5                   | 0.6779                  | $6e^{-3} $           | 0.6278         | $2e^{-3}$ |   |
> |   | Fully Finetuned 3B reward models | 3                   | 0.6973                  | $5e^{-4}$         | 0.6465         | $2e^{-3}$ |   |
>
> We observe that LoRA-based ensembles suffer from lower accuracy and higher variance compared to fully finetuned models, leading to worse performance.
>
> We thank the reviewer again for the comprehensive review to improve our paper. If the above rebuttal and additional experiments have addressed the reviewer's concerns, we would appreciate it if the reviewer could consider raising their score.

---

> ### Comment · Reviewer_2q4b · 2024-08-12
> **Replying Rebuttals**
>
> Thank you for your response. I appreciate the additional results provided by the authors. I would raise my score to 6.

---

### Official Review · Reviewer_JbJ9 · 2024-07-12

**Soundness:** 3
**Presentation:** 3
**Contribution:** 3
**Rating:** 6
**Confidence:** 3

**Summary:**

This paper studies the reward model overoptimization problem in RLHF. Specifically, they introduce a lightweight approach using adversarial policy optimization, provide corresponding justifications, and extensive empirical study to verify the proposed approach.

**Strengths:**

This paper studies important problems in RLHF, the solution is lightweight, and effectiveness is clearly demonstrated through experiments. The presentation is clear and the paper is easy to follow.

**Weaknesses:**

Please see the questions section.

**Questions:**

In the main text, it would be great to also show quantitative results on the uncertainty. To enhance the clarity, I would like to see two different types of presentations of the results:
1. (quantitative) correlation between the estimated uncertainty and the reward differences
2. (qualitative) a scatter plot showing the relationship between reward differences and estimated uncertainty (they are currently averaged in the figures)

How is the scalability of the proposed method? To be specific, the ensemble approaches may achieve better performance by using more RMs (regardless of the cost of doing so), and in the original paper scaling laws of RM overoptimization, another approach is to use larger RMs. How does the proposed method trade-off between cost and performance?

I would suggest the authors provide an algorithmic table of the proposed method to further enhance readability and clarity.

**Limitations:**

Please see the questions section.

---

> ### Author Rebuttal · Authors · 2024-08-07
>
> # Reply to Reviewer JbJ9
>
> First of all, we would like to thank the reviewer for encouraging comments as well as clarification questions. In particular, we would like to thank the reviewer for acknowledging that this paper “studies important problems in RLHF” and that the “effectiveness is clearly demonstrated through experiments”. In the following, we will clarify the remaining questions that the reviewer has mentioned in their review.
>
> > [Q1] "In the main text, it would be great to also show quantitative results on the uncertainty. To enhance the clarity, I would like to see two different types of presentations of the results: (1) (quantitative) correlation between the estimated uncertainty and the reward differences; (2) (qualitative) a scatter plot showing the relationship between reward differences and estimated uncertainty (they are currently averaged in the figures)"
>
> **Quantitative:**  We calculated the Pearson correlation between the estimated uncertainty and the reward differences. The results are as follows:
>
> | Dataset      | ENS-3B | ENS-7B | CI    |   |
> |--------------|--------|--------|-------|---|
> | Anthropic HH | 0.787  | 0.980  | 0.984 |   |
> | TLDR         | -0.594 | 0.909  | 0.994 |   |
>
> We can observe that CI achieves a similar Pearson correlation to ENS-7B, although the latter employs three 7B models. Furthermore, CI surpasses ENS-3B, a reward ensemble with comparable parameters.
>
> **Qualitative:** In terms of a qualitative scatter plot, we would like to mention that we have already added standard deviations in Figure 1, which we believe also demonstrates qualitatively the significance of the correlations between the reward differences and the uncertainty.
>
> > [Q2] "(1) Scalability of the proposed method and its trade-off between cost and performance. (2) Effect of the number of RMS and size of RM ensemble in ensemble-based approach."
>
> **For (1)**, We thank the reviewer for pointing out this interesting point. The proposed method aims to enhance RL optimization by leveraging lightweight uncertainty to mitigate overoptimization. It still falls under the umbrella of the RLHF pipeline, so it follows the general scalability of RLHF, meaning that larger policy or reward models will generally lead to better performance [1].
>
> On the other hand, our method does not require additional reward model training or maintaining multiple reward models in memory, making it highly scalable. Our method only requires precomputing and maintaining the matrix $M_D \in R^{d\times d}$, which summarizes all the last layer embeddings observed in the reward training data, where $d$ is the dimension of the last layer embedding in the reward model.
>
> **For (2)**, We found that simply scaling the number of ensembles without increasing the reward model size cannot improve its performance.  Specifically, we extended our analysis on uncertainty estimations in Figure 1 to include a configuration with five 3B ensembles, denoted as ENS-3B-5, given that accurate uncertainty estimation is a prerequisite for subsequent processes.
>
> The results are shown in Figure 1 in the attached PDF. We observed that ENS-3B-5 performs similarly to the three 3B ensembles (ENS-3B). Specifically, on the TLDR dataset, where ENS-3B showed poor performance as discussed in Section 5.2, ENS-3B-5 also failed to achieve reliable uncertainty estimation, as shown in Figure 1b in the attached PDF.
>
> This suggests that the size of the reward model might be more crucial than the number of ensembles. However, ensembling larger RMs incurs higher computational and memory costs. This again highlights the benefit of the proposed lightweight uncertainty estimation methods, which do not require additional memory. If feasible, larger reward models can be leveraged to boost RLHF performance.
>
> [1]Gao L, Schulman J, Hilton J. Scaling laws for reward model overoptimization, ICML2023
>
> > [Q3] " The algorithmic table of the proposed method."
>
> We thank the reviewer for this suggestion and have added the corresponding algorithmic table to the attached PDF.
>
> We hope that the above has addressed any outstanding questions and that the reviewer would consider raising their score if all the questions have been appropriately answered.

---

> > ### Comment · Reviewer_JbJ9 · 2024-08-12
> >
> > Thank you for the detailed response, I would like to keep my positive score.

---

### Author Rebuttal · Authors · 2024-08-07

# Common Response

> [CQ1] "Comparison with ensemble-based approach with reference response incorporated."

Following the reviewers' suggestions, we incorporated a variant of ENS-s, called ENS-ref, that leverages the same set of reference responses as our proposed method AdvPO.   More specifically, ENS-ref optimizes the following objective:
  $$\max\_{\pi_{\theta}} \mathbb{E}\_{x, y \sim \pi\_{\theta}(\cdot | x)} \left[ r\_{\rm ENS}(x,y)\right] - \mathbb{E}\_{x, y\_{\rm ref}}\left[ r\_{\rm ENS}(x,y_{\rm ref})\right] -\beta \mathbb{D}\_{\text{KL}}(\pi || \pi\_{\rm sft}).$$
where  $r\_{\rm ENS}(x,y)$ is  the reward of ENS-s, as defined in Eq. (3) in the paper.

We then follow the same experiment setup in Section 5.3 to perform RL optimization and compare the resulting policy with AdvPO. The results are as follows:
|             | Anthropic HH  | Anthropic HH  | Anthropic HH  | TLDR | TLDR | TLDR |
|-------------|---------------|---------------|---------------|------|------|------|
|             | **Win**           | **Tie**           | **Lose**          | **Win**  | **Tie**  | **Lose** |
| AdvPO v.s ENS-ref | 38%         | 40.5%        | 21.5%        | 76%   | 5%    | 19%   |

We can observe that AdvPO consistently outperforms ENS-ref on both datasets, even when the references have been added to ENS-s.

---

### Decision · Program_Chairs · 2024-09-25

**Decision:**

Accept (poster)

**Comment:**

The paper proposes a lightweight uncertainty quantification method to assess the reliability of the proxy reward and then maximize the pessimistic reward function within the predicted confidence ball. After some simplification and heuristic adjustments, the authors propose AdvPO, an optimization procedure to tackle the reward overoptimization problem in RLHF. Experiment results show that their method performs better than mean and Lora ensembles on both Anthropic HH and TL;DR datasets. Reviewers appreciate the lightweight method, clear math demonstration, and experiment design.